# Antinociceptive Interaction and Pharmacokinetics of the Combination Treatments of Methyleugenol Plus Diclofenac or Ketorolac

**DOI:** 10.3390/molecules25215106

**Published:** 2020-11-03

**Authors:** Héctor Isaac Rocha-González, María Elena Sánchez-Mendoza, Leticia Cruz-Antonio, Francisco Javier Flores-Murrieta, Xochilt Itzel Cornelio-Huerta, Jesús Arrieta

**Affiliations:** 1Escuela Superior de Medicina, Instituto Politécnico Nacional, Plan de San Luis y Díaz Mirón, Colonia Casco de Santo Tomás, Miguel Hidalgo, Ciudad de México 11340, Mexico; heisaac2013@hotmail.com (H.I.R.-G.); mesmendoza@hotmail.com (M.E.S.-M.); fjfloresmurrieta@yahoo.com.mx (F.J.F.-M.); xochilt_itz@hotmail.com (X.I.C.-H.); 2Facultad de Estudios Superiores Zaragoza, UNAM, Av. Guelatao No. 66, Colonia Ejército de Oriente, Iztapalapa, Ciudad de México 09230, Mexico; letycruza@yahoo.com.mx; 3Unidad de Investigación en Farmacología, Instituto Nacional de Enfermedades Respiratorias Ismael Cosió Villegas, Secretaría de Salud, Ciudad de México 14080, Mexico

**Keywords:** methyleugenol, isobolographic analysis, synergism, diclofenac, ketorolac

## Abstract

Although nonsteroidal anti-inflammatory drugs (NSAIDs) are one of the main types of drugs used to treat pain, they have several adverse effects, and such effects can be reduced by combining two analgesic drugs. The aim of this study was to evaluate the nociceptive activity of methyleugenol combined with either diclofenac or ketorolac, and determine certain parameters of pharmacokinetics. For the isobolographic analysis, the experimental effective dose 30 (ED_30_) was calculated for the drugs applied individually. With these effective doses, the peak plasma concentration (C_max_) was found and the other parameters of pharmacokinetics were established. Methyleugenol plus diclofenac and methyleugenol plus ketorolac decreased licking behavior in a dose-dependent manner in phase II, with an efficacy of 32.9 ± 9.3 and 39.8 ± 9.6%, respectively. According to the isobolographic analysis, the experimental and theoretical ED_30_ values were similar for methyleugenol plus diclofenac, suggesting an additive effect, but significantly different for methyleugenol plus ketorolac (3.6 ± 0.5 vs. 7.7 ± 0.6 mg/kg, respectively), indicating a probable synergistic interaction. Regarding pharmacokinetics, the only parameter showing a significant difference was C_max_ for the methyleugenol plus diclofenac combination. Even with this difference, the combinations studied may be advantageous for treating inflammatory pain, especially for the combination methyleugenol plus ketorolac.

## 1. Introduction

Pain is an unpleasant sensory and emotional experience associated with actual or potential tissue damage [1]. It functions as a defense mechanism to safeguard the integrity of the organism against potentially destructive factors. Under certain circumstances, however, pain does not provide a beneficial protective effect, but rather becomes a pathological process that requires treatment [2].

Pain includes motivational, emotional, discriminatory, sensory, affective, and cognitive aspects, which can lead to a low quality of life together with high social and economic costs [3]. Four main types of pain are currently recognized, classified by duration and the physiopathological characteristics: nociceptive, functional, neuropathic, and inflammatory [4].

Inflammatory pain results from tissue ruptures, intense pressures, burns, prolonged cold, and chemical injuries. A great variety of compounds are released by the injured cells, and still more are synthesized during post-injury events. Whereas some of these substances directly activate nociceptors, others such as prostaglandins sensitize them [5,6]. COX enzymes are responsible for the synthesis of prostaglandins [7]. The two isoforms of COX (COX1 and COX2) have a distinct distribution and tissue function. COX1 is expressed in basal conditions and synthesizes prostaglandins to perform homeostatic functions, while COX2 expression increases during inflammation and other disease states [8].

Non-steroidal anti-inflammatory drugs (NSAIDs) constitute a major class of analgesic drugs used to relieve inflammatory pain and inflammation through the inhibition of the COX enzymes [7]. Since selective inhibition of COX2 by new NSAIDs (e.g., celecoxib) blocks prostaglandin production at the sites of inflammation, gastric damage caused by the intake of this type of drug is minimal. However, these new NSAIDs have been found to damage the myocardium [9]. Traditional NSAIDs inhibit COX1 and COX2 to a greater or lesser extent, which is relevant in platelets and gastroduodenal mucosa because long-term inhibition can lead to ulcers with bleeding, perforation, or obstruction as well as renal dysfunction, cardiovascular events, and the risk of death [10]. Hence, the prolonged use of NSAIDs has limitations.

One alternative to decrease side effects is the combination of traditional NSAIDs with other types of analgesic drugs that act by a distinct mechanism of action. This allows a broader spectrum of pain relief activity and perhaps a synergistic effect, thereby reducing the individual doses and adverse effects of each drug [11]. One candidate for co-administration with NSAIDs is methyleugenol, a natural product known to have analgesic activity [12]. The aim of the present study was to evaluate (with mice and the formalin test) the analgesic effect of methyleugenol combined with either diclofenac or ketorolac, and determine (in rats) certain parameters of the pharmacokinetics of the two combination treatments at experimental effective dose 30 (ED_30_E).

## 2. Results

### 2.1. Antinociceptive Effect of the Individual Drugs

A subcutaneous 2.5% formalin injection into the right hindpaw of mice elicited a typical biphasic pattern of flinching behavior (Figure 1A). Oral administration of methyleugenol (Figure 1A,B), diclofenac (Figure 1C,D), and ketorolac (Figure 1E,F) significantly reduced licking behavior time in a dose-dependent fashion in phase II, but not phase I of the formalin test. According to the dose-response curve, methyleugenol reached an efficacy of 38.7 ± 3.9% at 30 mg/kg (Figure 1B) and had an ED_30_E of 8.4 ± 1.0 mg/kg (Table 1). Likewise, the experimental efficacy values of diclofenac and ketorolac were 30.1 ± 0.7% and 50.7 ± 7.5%, respectively, at 30 mg/kg (Figure 1D,F). In addition, the ED_30_ values for the two drugs were 31.6 ± 2.8 mg/kg and 7.1 ± 0.6 mg/kg, respectively (Table 1).

### 2.2. Antinociceptive Effect of the Drug Combinations

The combination of methyleugenol plus diclofenac at a 1:1 dose ratio (0.52 + 1.97, 1.05 + 3.95, 2.09 + 7.89, and 4.18 + 15.78 mg/kg) significantly decreased licking time with an efficacy value of 32.9 ± 9.3% (Figure 2A,C). The isobolographic analysis indicated that the ED_30_E and ED_30_T were similar (19.1 ± 3.3 mg/kg and 20.0 ± 1.5 mg/kg, respectively, Table 2), and that the confidence intervals of ED_30_E and ED_30_T overlapped (Figure 2E). Moreover, the interaction index was about 1 and the interaction index confidence interval crossed this value (Table 2). The statistical and isobolographic analyses evidence the likely additive interaction of the combined methyleugenol plus diclofenac treatment given to mice.

The combination of methyleugenol plus ketorolac at a 1:1 dose ratio (0.52 + 0.45, 1.05 + 0.89, 2.09 + 1.78, and 4.18 + 3.56 mg/kg) reduced licking time in a dose-dependent manner with an efficacy value of 39.8 ± 9.6% (Figure 2B,D). Interestingly, the ED_30_E of methyleugenol plus ketorolac (3.6 ± 0.5 mg/kg) was significantly lower than its ED_30_T (7.7 ± 0.6 mg/kg, Table 2). Furthermore, the confidence intervals of the ED_30_E and ED_30_T did not overlap (Figure 2F), the interaction index was less than 1, and the interaction index confidence interval did not pass through 1 (Table 2). Thus, statistical and isobolographic analyses suggest a synergistic antinociceptive interaction of the combined methyleugenol plus ketorolac treatment administered to mice.

### 2.3. Effect of Methyleugenol on the Pharmacokinetics of Diclofenac or Ketorolac

Regarding the pharmacokinetics of diclofenac (Table 3), the C_max_ for the individual treatment (2.80 ± 0.09 μg/mL) was significantly different from the value for methyleugenol plus diclofenac (1.83 ± 0.18 μg/mL). Hence, the absorption of diclofenac decreased with the combined treatment. However, there was no significant difference between these two treatments for the AUC_0→t_, AUC_0→∞_, t_1/2_, or T_max_ (Table 3).

Concerning the pharmacokinetics of ketorolac (Table 4), the C_max_ for the individual treatment (0.66 ± 0.06 µg/mL) was not significantly different from the value for methyleugenol plus ketorolac (0.97 ± 0.16 µg/mL). Thus, the absorption of ketorolac was not modified in the combined treatment. The other pharmacokinetic parameters did not differ significantly between these two treatments either (Table 4).

## 3. Discussion

Although NSAIDs are often prescribed to relieve pain and inflammation, their prolonged use has significant adverse effects [13]. One alternative to reduce such effects is to combine an NSAID with a prophylactic (e.g., a proton pump inhibitor). Another is the combination of two NSAIDs that act by distinct mechanisms of action. In the current contribution, the antinociceptive effect of the combination of an NSAID (diclofenac or ketorolac) with methyleugenol was examined on the formalin test. This test, based on the nociceptive response to chemical stimuli, is widely used to assess pain and preclinically evaluate analgesic drugs [14], due to its relative simplicity and high degree of reproducibility [15].

Methyleugenol, diclofenac, and ketorolac administered individually herein produced dose-dependent antinociception (Figure 1B,D,F), in agreement with previous reports [16,17,18]. The results showed significantly reduced licking behavior time in phase II of the test. This phase, involving both inflammatory mechanisms and central nervous system sensitization, is known to respond to various drugs with established clinical analgesic action (e.g., opiates, steroid or non-steroidal anti-inflammatory analgesics, N-methyl-D-aspartate antagonists, and gabapentin) [14]. Methyleugenol was more potent than diclofenac and equipotent with ketorolac, as indicated by the corresponding ED_30_ values (Table 1).

The combination of methyleugenol plus diclofenac demonstrated a dose-dependent effect (Figure 2C) and an additive-type antinociceptive interaction, the latter according to the isobolographic analysis (Figure 2E). Consequently, methyleugenol does not potentiate the effect of diclofenac. The additive interaction likely stems from the important shared signaling pathways of the antinociceptive activity of the two compounds including the inhibition of COX2 [12,19], the inactivation of NMDA [16,19], and the blocking of the upregulation of Nav1.7 [20,21].

The methyleugenol plus ketorolac combination showed a dose-dependent effect (Figure 2D) and a likely synergistic interaction, the latter evidenced by the isobolographic analysis (Figure 2F). Although ketorolac and methyleugenol have some common antinociceptive mechanisms of action, there are also substantial differences that could possibly explain the synergism in their interaction, whereas ketorolac inhibits COX1 and methyleugenol does not [12]; on the other hand, there are no reports that ketorolac blocks the upregulation of Nav1.7 channels or activates GABAA receptors and methyleugenol does.

Regarding the C_max_ for diclofenac, the study of pharmacokinetics revealed a significantly lower value with the combined treatment than with the administration of diclofenac alone (Table 3). In contrast, there was no significant difference in the C_max_ for methyleugenol plus ketorolac and ketorolac alone (Table 4). In both cases (methyleugenol plus diclofenac or ketorolac), the present results are in contrast with previously reported data by our work group [22], which demonstrated the inhibition of the absorption of diclofenac and ketorolac due to the combined treatment with methyleugenol. In the prior study, however, the dose of methyleugenol was higher (100 mg/kg vs. 3.99 mg/kg given currently), as was the dose of ketorolac (10 mg/kg vs. 1.64 mg/kg employed presently), while the dose of diclofenac was lower (10 mg/kg vs. 15.08 mg/kg administered currently).

The lower level of the C_max_ for diclofenac in the combination (versus individual) treatment was not a consequence of the presence of methyleugenol. Due to its lipophilic characteristics, methyleugenol has a rapid absorption [23] and thus does not modify gastric pH. As it does not generate acidic conditions, it does not alter the absorption of drugs [22] such as diclofenac. The reduced absorption of diclofenac is probably explained by its precipitation per se, considering its acidic characteristics [24] and the dose administered (15.08 mg/kg). This should cause a greater redissolution and a decrease in the precipitated fraction [25,26]. The rate of dissolution of a drug in the gastrointestinal tract often partially or completely controls the rate of its absorption [27]. Regarding T_max_, t_1/2_ and the AUC, no significant differences were observed between groups (Table 3). Therefore, the additive interaction between methyleugenol and diclofenac cannot be attributed to the pharmacokinetic interaction. In the same sense, the parameters of t_1/2_, T_max_, AUC, and C_max_ (Table 4) showed no significant difference between methyleugenol plus ketorolac and ketorolac alone, which indicates that methyleugenol does not interfere with the absorption of ketorolac.

The differences in the mechanisms of action of methyleugenol and ketorolac could explain the apparent synergistic interaction involved in the combined treatment. Likewise, the similarity in the mechanisms of action of methyleugenol and diclofenac likely accounts for the additive effect of methyleugenol and diclofenac in the combined treatment.

## 4. Material and Methods

### 4.1. Animals

Male ICR mice (20–25 g) and Wistar rats (180–220 g) were obtained from our breeding facilities. The animals were housed in a vivarium under controlled conditions, with the temperature at 22–25 °C, air flow, a 12-h light/dark cycle, and free access to food and water. They were fasted for 18 h before the experiments. The care and handling of animals were conducted in accordance with the Mexican official guidelines for laboratory animals (NOM-062-ZOO-1999) [28] and the “Ethical guidelines for investigations of experimental pain in conscious animals” [29]. In addition, all animal experiments were approved by the institutional Ethics in Research Committee. Every effort was made to minimize the pain and suffering of the animals, utilizing the minimum number of animals for the statistical power needed to find a significant effect. Each assay was carried out with independent groups of animals, used only once, and euthanized in a CO_2_ chamber at the end of the experiment.

### 4.2. Drugs

Formaldehyde, ketorolac, diclofenac, and methyleugenol were purchased from Sigma-Aldrich (St. Louis, MO, USA). Methanol and high-performance liquid chromatography (HPLC)-grade acetonitrile were acquired from JT Baker. Formaldehyde was freshly prepared in distilled water. Ketorolac and diclofenac were dissolved in saline solution (0.9%), while methyleugenol was suspended in 0.05% Tween 80.

### 4.3. Formalin Test

Mice were orally administered one of six treatments (0.1 mL/10 g): (1) the vehicle, a saline solution with 0.05% Tween 80 (the control); (2) methyleugenol (1–30 mg/kg); (3) diclofenac (1–30 mg/kg); (4) ketorolac (1–30 mg/kg); (5) methyleugenol plus diclofenac; and (6) methyleugenol plus ketorolac. The 2.5% formalin test was conducted 30 min after the animals received each treatment. Formalin-induced licking behavior in mice was evaluated as previously described [30]. Briefly, each mouse was placed in an open plastic observation chamber for 30 min to become acclimated to its surroundings. Subsequently, it was removed, injected with 20 μL of 2.5% formalin into the dorsum of the right hindpaw, and returned to the chamber. The accumulated time spent licking the injected paw was taken as nociceptive behavior. Animal behavior was observed during phase I (from 1–10 min) and phase II (from 11–40 min). A brief timeline of the experimental design is given in Figure 3.

#### 4.3.1. Data Analysis from the Formalin Test

The results of this assay are expressed as the mean ± standard error of the mean (SEM) of 6–8 animals. The curve for the time course of the activity of each drug dose was constructed by plotting the licking time against the log dose. The percentage of the antinociceptive effect was calculated from the total licking time evoked during phase II, in accordance with the following equation:$$\%\;\mathrm{Antinociception}\;=\;\frac{\mathrm{Control}\;\mathrm{licking}\;\mathrm{time}\;-\;\mathrm{Test}\;\mathrm{licking}\;\mathrm{time}}{\mathrm{Control}\;\mathrm{licking}\;\mathrm{time}}\;\times \;100$$

The statistical differences between groups with regard to the dose-response curves were determined by one-way analysis of variance (ANOVA) followed by Dunnett’s test.

#### 4.3.2. Isobolographic Analysis

The ED_30_ for each drug administered individual was calculated from its dose-response curve by linear regression. The isobologram was then constructed by plotting the ED_30_ value of methyleugenol on the abscissa and the ED_30_ value of diclofenac or ketorolac on the ordinate to obtain the theoretical additive line. The theoretical ED_30_T value for each combination was calculated and the dose-response curve was constructed based on fractions (1/2, 1⁄4, 1⁄8m and 1⁄16) of the ED_30_ values of the individual drugs, using a 1:1 dose ratio. Afterward, the experimental ED_30_E value of each combination was calculated from its corresponding dose-response curve by linear regression. The difference between each ED_30_T and the respective ED_30_E was examined by the Student’s t-test. Interaction indices (γ) and confidence intervals for the ED_30_ were calculated as described by Tallarida [31,32].

### 4.4. Evaluation of Pharmacokinetics

The parameters of the pharmacokinetics of diclofenac and ketorolac were ascertained in four groups of rats. Two groups (the controls) received 0.05% Tween 80, and 30 min later, 15.08 mg/kg of diclofenac or 1.64 mg/kg of ketorolac. The other two groups received methyleugenol at a dose of 3.99 or 1.9 mg/kgm and 30 min later 15.08 mg/kg of diclofenac or 1.64 mg/kg of ketorolac, respectively. All treatments were administered orally and in a volume of 0.5 mL/100 g. Blood samples of 200 microliters were drawn as previously reported [22]. Briefly, the animals were subjected to cannulation of the caudal artery, employing a PE-10 catheter (rinsed with heparin) to extract 200 microliters of blood. The samples were taken at 2.5, 5.7, 5, 10, 15, 30, 45, 60, 120, 240, and 360 min post-administration of diclofenac, and at 0, 5, 15, 30, 45, 60, 120, 180, 240, and 360 min post-delivery of ketorolac. Immediately after drawing a blood sample, the same volume of physiological isotonic saline was injected to avoid hypovolemia. Each sample was centrifuged, frozen, and left in cold storage to await further use. A brief timeline of the experimental design is illustrated in Figure 3.

#### 4.4.1. Quantifying the Plasma Concentration of Diclofenac and Ketorolac

The drugs were assayed by HPLC on a Waters system to assess the plasma concentration of diclofenac and ketorolac, as previously described (with slight modifications) [22]. Briefly, reversed phase chromatography was used for drug analysis. Ketorolac was examined in a Zorbax Eclipse Plus C18 column (150 × 4.6 mm, 4.5 μm) using a mobile phase of 0.04 M phosphate buffer:acetonitrile:methanol 60:20:20 (*v*/*v*/*v*) delivered at a flow rate of 1.4 mL/min. Diclofenac was evaluated in a Symmetry C18 Waters column (150 × 4.6 mm, 5 µm), with a mobile phase of 0.041 M phosphate buffer:acetonitrile:methanol 49:51 (*v*/*v*) at a flow rate of 1.4 mL/min. The lower limit of quantification was 0.5 µg/mL for diclofenac and 0.05 µg/mL for ketorolac. For diclofenac, the intra- and inter-day precision coefficients of variation (CV) were less than 3.86% and 5.97%, respectively. The accuracy of the intra- and inter-day determinations was 93.62–100.82% and 95.80–99.8%, respectively. For ketorolac, the intra- and inter-day precision CV were less than 8.45% and 10.03%, respectively. The accuracy of the intra- and inter-day determinations was 96.87–119.14% and 93.75–108.3%, respectively.

#### 4.4.2. Non-Compartmental Analysis

The non-compartmental pharmacokinetic parameters, estimated by Phoenix^®^ (WinNonlin^®^ ver 8.1), were the terminal half-life (t_1/2_), peak plasma concentration (C_max_), time to reach C_max_ (T_max_), and area under the plasma concentration vs. time curve from time zero to the last observation time (AUC_0→t_) and from time zero to infinity (AUC_0→∞_). Data were expressed as the mean ± SEM and examined with unpaired t-tests for comparisons between two means, considering statistical significance at *p* < 0.05.

## 5. Conclusions

The results of the evaluation of the analgesic activity of diclofenac, ketorolac, and methyleugenol administered individually coincide with the reported data. The combination of methyleugenol and ketorolac, according to the current findings, likely produces a synergistic effect. In such a case, it should be useful in the treatment of inflammatory pain. In contrast, the combination of methyleugenol and diclofenac had an additive effect, whereas the mechanisms of action of methyleugenol and diclofenac are very similar, and those of methyleugenol and ketorolac exhibit important differences, which is probably the best explanation for the aforementioned drug interactions. Further research is needed to provide greater clarity about the mechanisms involved in the interactions of the compounds in these combinations.

## Figures and Tables

**Figure 1 molecules-25-05106-f001:**
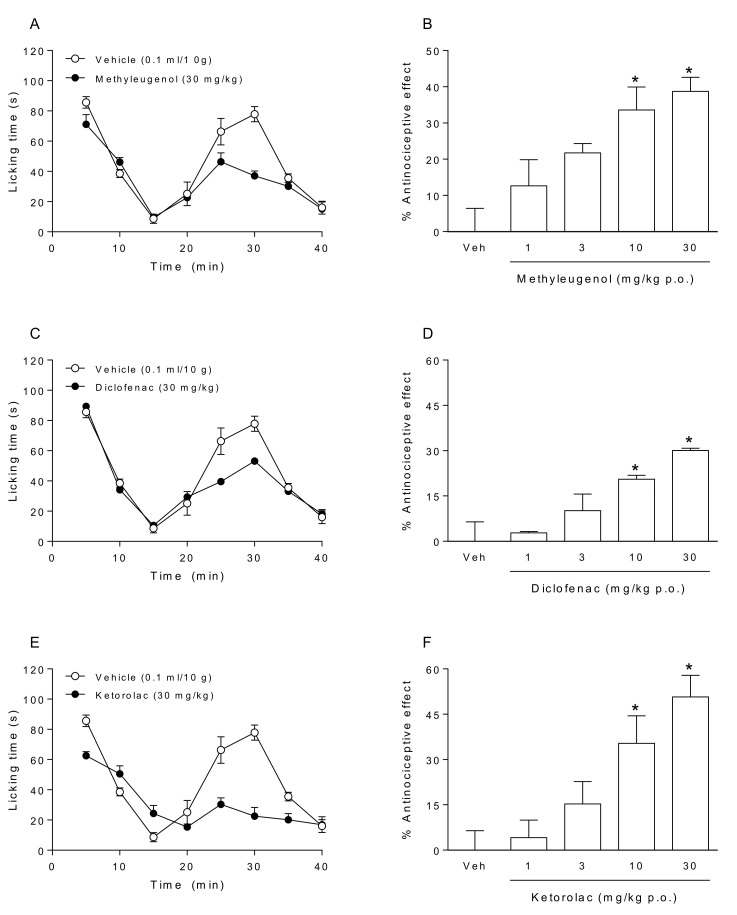
The time course of the antinociceptive effect found for rats after receiving each compound individually: (**A**) methyleugenol (30 mg/kg), (**C**) diclofenac (30 mg/kg), and (**E**) ketorolac (30 mg/kg). Bar graph of the dose-response effect on rats produced by the treatment with the vehicle (Veh), (**B**) methyleugenol, (**D**) diclofenac, or (**F**) ketorolac. The data reflect the results of phase II of the formalin test. Bars depict the mean of the percentage of antinociception ± SEM for 6–8 animals. * *p* < 0.05 versus Veh group, as established by one-way analysis of variance (ANOVA) followed by Tukey’s test.

**Figure 2 molecules-25-05106-f002:**
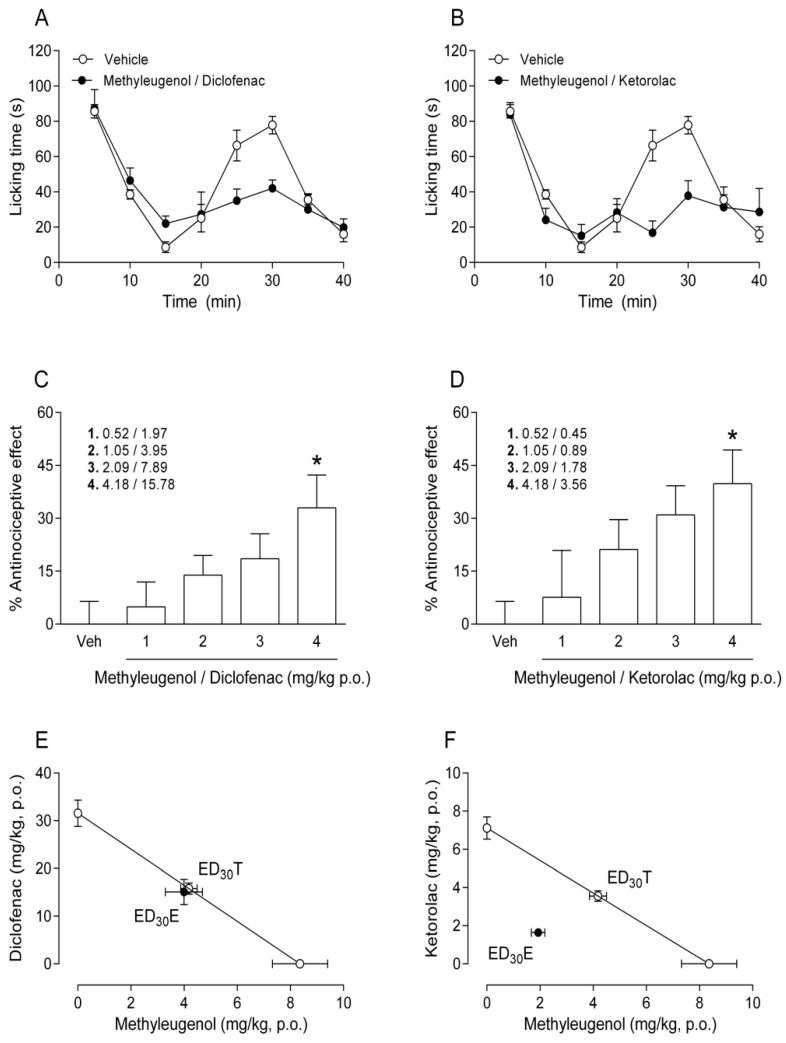
The time course of the antinociceptive effect induced by methyleugenol plus diclofenac (**A**) or methyleugenol plus ketorolac (**B**), evaluated by the formalin test. The dose-response effects of the vehicle (Veh), methyleugenol plus diclofenac (**C**), and methyleugenol plus ketorolac (**D**). Bars illustrate the mean of the percentage of antinociception ± SEM for 6–8 animals. * *p* < 0.05 versus the Veh group, determined by one-way analysis of variance (ANOVA) followed by Tukey’s test. The isobolographic interaction of methyleugenol plus diclofenac (**E**) or methyleugenol plus ketorolac (**F**) at a 1:1 dose ratio. The points on the *X*-axis portray the experimental ED_30_ values of methyleugenol and those on the *Y*-axis the experimental values of diclofenac or ketorolac. The diagonal line connecting the ED_30_ of the combination of methyleugenol and diclofenac or ketorolac is the theoretical value of additivity. For each combination, the point designated as ED_30_T represents the theoretical ED_30_, and the point labelled ED_30_E indicates the experimental ED_30_ for each combination. The ED_30_E was statistically different from the ED_30_T for methyleugenol plus ketorolac but not for methyleugenol plus diclofenac. Differences were examined with the Student’s *t*-test, considering significance at *p* < 0.05.

**Figure 3 molecules-25-05106-f003:**
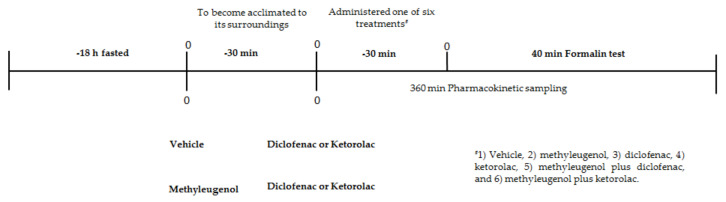
Timeline of the experimental design.

**Table 1 molecules-25-05106-t001:** Antinociceptive effective dose 30 (ED_30_) were derived from the formalin test applied to mice. Data were established by linear regression of the dose-response curves of methyleugenol, diclofenac, and ketorolac when administered alone.

Drug	ED_30_ ± SEM (mg/kg)	R^2^
Methyleugenol	8.4 ± 1.0	0.983
Diclofenac	31.6 ± 2.8	0.997
Ketorolac	7.1 ± 0.6	0.992

Abbreviations: SEM = Standard error of the mean, R^2^ = correlation coefficient.

**Table 2 molecules-25-05106-t002:** Statistical analysis of the effects produced on mice by the administration of methyleugenol plus diclofenac and methyleugenol plus ketorolac. The experimental values were determined by the formalin test.

Dose Ratio	Theoretical ED_30_ ± SEM (CI at 90%)	Experimental ED_30_ ± SEM (CI at 90%)	γ ± SEM (CI at 90%)
methyleugenol + diclofenac 1:1	20.0 ± 1.5 (16.2 − 24.5)	19.1 ± 3.3 (9.0 − 40.5)	0.96 ± 0.18 (0.63 − 1.46)
methyleugenol + ketorolac 1:1	7.7 ± 0.6 (6.3 − 9.6)	3.6 ± 0.5 * (2.0 − 6.2)	0.46 ± 0.07 (0.33 − 0.65)

* Statistically different from the theoretical ED_30_ (*p* < 0.05), evaluated by the Student’s *t*-test. Abbreviations: SEM = standard error of the mean; ED_30_ = effective dose 30 (in mg/kg); CI = confidence interval at 90%; γ = interaction index.

**Table 3 molecules-25-05106-t003:** Parameters of pharmacokinetics determined after the oral administration of the vehicle plus diclofenac (15.08 mg/kg) or methyleugenol plus diclofenac (3.99 and 15.08 mg/kg, respectively) to rats. Data are expressed as the mean ± SEM (*n* = 6).

Parameter	Diclofenac	Methyleugenol + Diclofenac
C_max_ (µg/mL)	2.80 ± 0.09	1.83 ± 0.18 *
t_½_ (min)	126.71 ± 22.25	103.87 ± 12.11
T_max_ (min)	6.00 ± 0.55	10.83 ± 2.00
AUC_0→t_ (µg × min/mL)	149.02 ± 30.83	158.76 ± 19.83
AUC_0→∞_ (µg × min/mL)	191.63 ± 49.50	189.05 ± 27.39

* *p* ≤ 0.05 compared to the control group; unpaired *t*-tests.

**Table 4 molecules-25-05106-t004:** Parameters of the pharmacokinetics ascertained after oral administration of the vehicle plus ketorolac (1.64 mg/kg) or methyleugenol plus ketorolac (1.9 and 1.64 mg/kg, respectively) to rats. Data represent the mean ± SEM (*n* = 6).

Parameter	Ketorolac	Methyleugenol + Ketorolac
C_max_ (µg/mL)	0.66 ± 0.06	0.97 ± 0.16
t_½_ (min)	39.45 ± 5.45	44.82 ± 5.79
T_max_ (min)	13.33 ± 5.27	10.00 ± 2.33
AUC_0→t_ (µg × min/mL)	39.77 ± 4.01	53.13 ± 6.99
AUC_0→∞_ (µg × min/mL)	41.29 ± 4.51	56.34 ± 7.09

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
