# Peer review of "Antinociceptive Interaction and Pharmacokinetics of the Combination Treatments of Methyleugenol Plus Diclofenac or Ketorolac"

_molecules, 2020, doi:10.3390/molecules25215106_

Round 1

Reviewer 1 Report

I have the pleasure to review this interesting manuscript titled: “Antinociceptive interaction and pharmacokinetics of the combination treatments of methyleugenol plus diclofenac or ketorolac”. This manuscript seems interesting and promising results in analgesic and anti-inflammatory activity, as well as possible organ toxicity of methyleugenol combined with either diclofenac or  ketorolac. The aim of this  study was to evaluate the nociceptive activity  and determine certain parameters of pharmacokinetics.

This manuscript is very well written. However, the idea to show the different parameters of pharmacokinetics develop as a new treatment option for pain. However, this is new and innovating and the question is relevant in daily practice.

Specific comments

I reviewed this interesting manuscript.  

This manuscript seems interesting and promising results in a rather special area of NARS. The results of a small sample size demonstrated good results.

Methods

Please describe the section of methods

Author Response

  1. Extensive editing of English language and style required

The manuscript has been thoroughly proofread by a native English speaker familiar with the field.

  1. Methods

Please describe the section of methods

Response: Thank you for your observation, which has been taken into account

Reviewer 2 Report

The manuscript by Rocha-Gonzálezet al ” Antinociceptive interaction and pharmacokinetics of the combination treatments of methyleugenol plus diclofenac or ketorolac”

The manuscript is interesting in its field, showing an overview of the pharmacokinetics in formalin induced pain model.

I recommend minor revision for the paper:

  • The authors should improve the english language

  • The author should add the timeline of the experimental design and specify the time of tissue and plasma collection

  • Why did the author choose the formalin test as the pain model? The author should argue this part in the discussion

  • Figure 3 there is an error in the legend (Diclofena)

  • The author should update the bibliography in the introduction the author should add recent paper published regarding the association between anti inflammatory compound and NSAID(for example doi: 10.3390/ijms21103509)

Author Response

  1. The authors should improve the English language

Response: The manuscript has been thoroughly proofread by a native English speaker familiar with the field.

  1. The author should add the timeline of the experimental design and specify the time of tissue and plasma collection.

Response: The timeline has been included in the text.

  1. Why did the author choose the formalin test as the pain model? The author should argue this part in the discussion.

Response: Thank you for your observation, which has been taken into account.

  1. Figure 3 there is an error in the legend (Diclofena).

Response: Thanks for your observation, the error has been corrected.

  1. The author should update the bibliography in the introduction the author should add recent paper published regarding the association between anti-inflammatory compound and NSAID(for example doi: 10.3390/ijms21103509).

Response: The references of these sections have been enriched.